# Prospective Associations of Physical Activity and Health-Related Physical Fitness in Adolescents with Down Syndrome: The UP&DOWN Longitudinal Study

**DOI:** 10.3390/ijerph18115521

**Published:** 2021-05-21

**Authors:** Borja Suarez-Villadat, Ariel Villagra, Oscar L. Veiga, Veronica Cabanas-Sanchez, Rocio Izquierdo-Gomez

**Affiliations:** 1Department of Physical Activity and Sport Sciences, Alfonso X el Sabio University, 28691 Madrid, Spain; 2Department of Physical Education, Sport and Human Movement, Autonomous University of Madrid, 28049 Madrid, Spain; ariel.villagra@uam.es (A.V.); oscar.veiga@uam.es (O.L.V.); 3IMDEA Food Institute, CEI UAM+CSIC, 28049 Madrid, Spain; veronica.cabanas@imdea.org; 4GALENO Research Group, Department of Physical Education, Faculty of Education Sciences, University of Cádiz, Avenida República Saharaui, Puerto Real, 11519 Cádiz, Spain; rocio.izquierdo@uca.es; 5Instituto de Investigación e Innovación Biomédica de Cádiz (INiBICA), 11009 Cádiz, Spain

**Keywords:** physical fitness, longitudinal study, accelerometer-based PA, Down syndrome

## Abstract

(1) Background: Numerous studies have focused on examining the association between PA levels and health-related physical fitness components in children or adolescents without disabilities. However, research on the association between PA and health-related physical fitness in adolescents with DS (Down syndrome) is limited, and most of the previous studies have been developed with a cross-sectional perspective. Therefore, the aim of the present study was to assess the prospective association of accelerometer-based PA at baseline with health-related physical fitness at a 2-year follow-up in a relatively large sample of adolescents with DS from the UP&DOWN study. (2) Methods: A total of 92 adolescents with DS (58 males) between 11 and 20 years old with full data were eligible from an initial sample of 110 participants. Fitness was assessed by the ALPHA health-related fitness test battery for youth, and physical activity was assessed by Actigraph accelerometers. (3) Results: The high tertile of total PA was related to decreased motor (Beta [95% CI] = −1.46 [−2.88; −0.05]) and cardiorespiratory fitness (Beta [95% CI] = −2.22 [−4.42; 0.02]) in adolescents with DS. (4) Conclusions: In adolescents with DS, (i) PA level was not prospectively associated with muscular fitness and (ii) high levels of total PA at the baseline were inversely associated with motor and cardiorespiratory fitness at the 2-year follow-up. For comparative purposes, these relationships were also examined in a subsample of adolescents without DS.

## 1. Introduction

Physical activity (PA) could be defined as any body movement produced by the action of skeletal muscles that is capable of causing an increase in energy expenditure [1]. It is considered to be the cornerstone for the maintenance and development of healthy lifestyle habits in the adolescent population [2]. Physical fitness, the main components of which are cardiorespiratory, musculoskeletal, and motor fitness, is considered a powerful marker of health during youth [3]. PA levels are commonly related to adolescents’ physical fitness, but this assumption is not obvious in the population with Down syndrome (DS).

DS is a genetic condition associated with mild or moderate intellectual disability. It is caused by the presence of additional genetic material known as 21 trisomy, with an estimated incidence of 1 in 1000 live births [4]. Generally, people with DS have motor and developmental delays, exhibiting common phenotypic characteristics, such as diminished cardiac responses, muscle hypotonia, blunted arterial stiffness responses, poor postural control, autonomic dysfunction, chronotropic incompetence, and poor balance. This might be the cause of low levels of PA and health-related physical fitness [5,6].

To date, numerous studies have focused on examining the association between PA levels and health-related physical fitness components in children or adolescents without disabilities. For example, cross-sectional studies showed that moderate PA (MPA), vigorous PA (VPA), or moderate-vigorous PA (MVPA) were positively related with cardiorespiratory fitness [7,8], muscular fitness [9], and motor fitness [2]. Moreover, longitudinal analyses among adolescents without disabilities reported positive associations of VPA with muscular fitness [10] and cardiorespiratory fitness [11]. However, research on the association between PA and health-related physical fitness in adolescents with DS is limited, and most of the previous studies were developed with a cross-sectional perspective [12,13,14]. For example, Izquierdo-Gomez et al. [12] showed that high levels of PA, particularly VPA, were positively associated with muscular fitness, motor fitness, and cardiorespiratory fitness; Matute-Llorente et al. [13] found that MPA and MVPA correlated with cardiorespiratory fitness in adolescents with DS. However, the study conducted by Shields et al. [14] in a group of children with DS of 7–17 years showed that there was no relationship between the levels of physical activity and the cardiorespiratory capacity of this population.

Nevertheless, prospective studies aimed at analyzing the longitudinal relationship between PA and health-related physical fitness in youth with DS are needed. 

Therefore, the aim of the present study was to assess the prospective association of accelerometer-based PA at baseline with health-related physical fitness (i.e., muscular fitness, motor fitness, and cardiorespiratory fitness) at a 2-year follow-up in a relatively large sample of adolescents with DS from the UP&DOWN study. For comparative purposes, these relationships were also examined in a subsample of adolescents without DS. 

## 2. Materials and Methods

### 2.1. Study Design and Sample

The present study was carried out in relation to the UP&DOWN study (“Follow-up in healthy schoolchildren and in adolescents with DOWN syndrome: psycho-environmental and genetic determinants of physical activity and its impact on fitness, cardiovascular diseases, inflammatory biomarkers and mental health; the UP&DOWN Study”). Full methodological details have been described elsewhere [15]. Briefly, the UP&DOWN study was developed in 3 phases: (i) the baseline data collection was performed between 2011 and 2012, (ii) the first follow-up was made between 2012 and 2013, and (iii) the second follow-up was completed between 2013 and 2014. For the present study, data from the baseline and second follow-up were used. 

A total of 92 adolescents with DS (58 males) between 11 and 20 years old with full data were eligible from an initial sample of 110 participants. We also included a sex-matched non-DS group of 155 (105 males) participants with full data randomly selected from 673 adolescents without DS, aged from 12 to 18 years old, recruited into the UP&DOWN study. Adolescents with DS were recruited from associations and foundations for people with intellectual disabilities and special education schools (15 centers) in Madrid and Toledo (Spain), while adolescents without DS were recruited from schools in Madrid (Spain). All participants with DS had to accomplish two requirements to be included in the study: (i) to have an intelligence quotient above 35; (ii) to be able to perform a physical fitness test without having a motor disability that could affect the results (e.g., wheelchair users, those with injuries), and (iii) not to have heart diseases (e.g., coronary artery disease, congenital heart defects).

Participants and parents or guardians were informed about the purpose of the study through written letters, and signed consent was required to participate. The study protocol was approved by the Ethics Committee of the Hospital Puerta de Hierro (Madrid, Spain) and the Bioethics Committee of the National Research Council (Madrid, Spain).

### 2.2. Assessment of Health-Related Physical Fitness

Fitness was assessed by the ALPHA (Assessing Level of Physical Activity) health-related fitness test battery for youth [16,17]. More detailed descriptions of each test in adolescents with or without DS have been reported elsewhere [18].

Muscular fitness was computed from the handgrip strength test and the standing long jump test. The isometric muscular strength of upper limbs was measured using a handheld dynamometer with adjustable grip (TKK 5101 Grip D, Takey, Tokyo, Japan). Participants with DS gradually and continuously depressed the dynamometer for at least 2 s in a seated position, whereas the standing position was used for adolescents without DS. No significant differences in handgrip strength scores between positions (seated vs. standing) were found in adolescents without DS in the pilot study [18]. The highest score for each hand was recorded, and then the mean scores for both hands were calculated. The explosive muscular strength of lower limbs was assessed by the standing long jump test. The participant, beginning from a starting position behind the line and with feet together, was to jump as far as possible until landing with both feet together. The best attempt was recorded. The individual scores of the handgrip strength and standing long jump tests were standardized (i.e., [value-mean]/standard deviation). Finally, the muscular fitness score was calculated as the average of the standardized values of both tests [19]. 

Motor fitness was measured using the 4 × 10 m shuttle-run test. Participants had to run four times at maximum speed between two parallel lines 10 m apart, exchanging the sponges placed on the ground. The time spent in completing the test was recorded in seconds, but it was multiplied by −1 in order to achieve higher scores indicating better motor fitness.

Cardiorespiratory fitness was assessed by using the 20 m shuttle-run test. Participants had to run in a straight line between two lines 20 m apart and pivot when completing a shuttle, following a rhythm determined by a recorded audio CD. The test started with a speed of 8.5 km h^−1^, which increased 0.5 km h^−1^ every minute (1 min = one stage). Adolescents were instructed to run while maintaining the rhythm imposed for as long as possible, and the test was finished when the participant did not reach the line on two consecutive occasions or when voluntarily withdrawing due to excessive fatigue. The test score was recorded as the number of laps (i.e., 20 m) completed. 

All physical fitness tests of the ALPHA health-related fitness test battery were performed twice, except the cardiorespiratory fitness test, which was performed only once. To solve possible comprehension difficulties, the researchers performed the tests together with the adolescents with DS (one by one), praising the correct form of execution [18].

### 2.3. Assessment of PA

PA was assessed by Actigraph accelerometers, models GT1M, GT3X, and GT3X+ (ActigraphTM, Pensacola, FL, USA). The epoch was set at 2 s (for GT1M models) and at 30 Hz (for GT3X and GT3X+ models), but all data were reintegrated into a 10 s epoch before analyses [20]. The device was located at the lower back, and participants were required to wear the accelerometer for 7 consecutive days, removing it only for sleep and water-based activities. Adolescents had to have at least 3 days of valid data, with a minimum of 8 h of recording per day, in order to be included in the study [20]. 

Nonwear time was identified as a period of 60 min of zero counts, considering an allowance window of up to two consecutive minutes of <100 counts per minute (cpm) with the up/downstream of 30 consecutive minutes of zero counts for the detection of artifactual movements [21]. Variables included in the present analyses were MPA, VPA, and MVPA expressed in minutes per day (min/day) and total accelerometer-based PA (Total PA) expressed in count per minute (cpm). MVPA was regarded as the sum of MPA and VPA. The cutoff points developed by the Helena study [17] were used to evaluate the intensities of MPA and VPA. These cutoff points to define levels of PA are similar to those used in previous studies with European children and adolescents [22]. 

### 2.4. Covariates

#### 2.4.1. Assessment of Body Mass Index (BMI)

Weight and height were measured in participants without shoes and light clothing, following standard procedures [17,18]. Weight was assessed with an electronic scale (model SECA 701, Hamburg, Germany) with an accuracy of 0.1 kg. Height was obtained using a telescopic measuring instrument (model SECA 701, Hamburg, Germany) to the nearest 1 mm. BMI was calculated by weight in kilograms divided by squared height, in meters.

#### 2.4.2. Assessment of Adaptive Behavior

The adaptive behavior questionnaire was used to assess the functional profile of adolescents with DS. This specific questionnaire was designed by our research group based on the Adaptive School Scale assessments (ABS-S:2) [23]. Prior to the study, a total of 22 families with adolescents with intellectual disabilities, including DS, were selected to check the reliability of the questionnaire. Results showed a high internal consistency (α = 0.87) and a one-week test–retest reliability (intraclass correlation coefficient = 0.88; *p* < 0.001) for the total score. This questionnaire was composed of 12 items about autonomous, routine, and social skills scored on four response categories (1 = none; 2 = little; 3 = fairly; and 4 = a lot). The final score was calculated by adding all items, with higher scores indicating better adaptive behavior.

### 2.5. Statistical Analysis

All statistical analyses were performed by using the IBM SPSS statistical software package (v.24.0, Chicago, IL, USA) for Macintosh, and the level of significance was set at *p* < 0.05. The descriptive characteristics of the study sample were presented as means and standard deviations (SD). Participants were classified using tertiles of PA levels at baseline (low, middle, high). Linear regression analysis was performed to analyze the prospective association of PA levels (predictor variables; low tertile as reference category) at baseline with health-related physical fitness variables (outcomes) at the 2-year follow-up using 2 separate models. Model 1 was adjusted for sex and age (years). Model 2 was adjusted for variables in model 1 plus BMI at baseline and the corresponding baseline value of health-related physical fitness. Moreover, an adaptive behavior value was included as a covariate in model 2 for the DS group. Additionally, we examined whether meeting the international recommendation of PA [24] at baseline was associated with health-related physical fitness variables at the 2-year follow-up (<60 min/day and >60 min/day of MVPA).

## 3. Results

Table 1 shows descriptive characteristics of the study sample at the baseline and 2-year follow-up in adolescents with and without DS. The group with DS shows an increase in BMI and a better performance of cardiorespiratory fitness levels (laps) at the 2-year follow-up, while the group without DS shows an increase in BMI and a better performance of motor fitness and cardiorespiratory fitness variable. However, total PA and MPA decreased significantly at the 2-year follow-up.

Table 2 displays the association of tertiles of PA at baseline with health-related physical fitness at the 2-year follow-up in adolescents with and without DS. In fully adjusted models for adolescents with DS, compared with the reference category (i.e., low tertile of total PA), the high tertile of total PA was related to decreased motor (Beta [95% CI] = −1.46 [−2.88; −0.05]) and cardiorespiratory fitness (Beta [95% CI] = −2.22 [−4.42; 0.02]). On the contrary, among adolescents without DS, compared to the corresponding reference categories, the higher tertile of MPA and the middle tertile of MVPA were associated with increased muscular (Beta [95% CI] = 0.44 [0.13; 0.75]) and cardiorespiratory fitness (Beta [95% CI] = 6.39 [0.07; 12.70]), respectively. Finally, accomplishments with the international recommendation of PA at baseline were not associated with health-related physical fitness variables at the 2-year follow-up (data not shown).

## 4. Discussion

To the best of our knowledge, this is the first study to examine prospective associations between accelerometer-measured PA with health-related physical fitness in a relatively large sample of adolescents with DS. The main findings suggest that the total PA at baseline was inversely associated with motor and cardiorespiratory fitness at the 2-year follow-up in adolescents with DS. On the other hand, in adolescents without DS, MPA and MVPA at baseline were positively associated with muscular and cardiorespiratory fitness at the 2-year follow-up. The results suggest that the healthy relationship between PA levels and health-related physical fitness found in young people without disabilities is not evident among adolescents with DS. These results contribute to the current scientific literature in adolescents with DS, but future longitudinal studies are needed to understand the association of PA and health-related physical fitness in this population. A previous study in adolescents without disabilities produced similar results to the association between accelerometer-measured VPA and muscular fitness in our study [10]. These authors suggested that VPA at baseline was significantly associated with the muscular fitness variable at a 24-month follow-up in boys but not in girls. Similarly, our study showed that only those adolescents located in the higher tertile of MPA at baseline showed a significant increase in muscular fitness at the 2-year follow-up. Concerning adolescents with DS, the findings observed in our study are discordant with those of previous cross-sectional studies with and without DS [12]. This study reported that those adolescents with DS with higher levels of total PA, VPA and MVPA had higher values in muscular fitness, observing a positive relationship [12]. One of the possible reasons why there is no relation between different levels of physical activity and the force variable may be due to the development of this capacity in the adolescent population with DS. The lower muscle strength could be related to physiological characteristics and a combination of low levels of physical activity and high sedentariness [4,25].

Regarding motor fitness, some longitudinal studies examined the association between PA and motor fitness in adolescents without disability. For example, Jaakkola et al. [11], in a prospective study, showed a positive association between MVPA and motor fitness in adolescents. Also, Lima et al. [26] found a positive reciprocal longitudinal relationship across childhood and early adolescence between VPA or MVPA and motor fitness. However, our results suggested that there were no association between PA level at baseline and motor fitness at 2-years follow-up in adolescents without DS. Concerning adolescents with DS, cross-sectional studies in this populations showed that total PA and VPA levels were positive associated with motor fitness [12]. Instead, our results with longitudinal data showed that those adolescents located in high tertiles of total PA performance a worsened motor fitness after 2-years follow-up.

With respect cardiorespiratory fitness, there is a strong cross-sectional and longitudinal evidence suggesting that high levels of PA, especially VPA, are a key to improve cardiorespiratory fitness in adolescents without disability [27]. Findings of the present study in adolescents without DS are in accordance with the scientific literature showing that MVPA levels was associated with an improvement of cardiorespiratory fitness at 2-years follow-up. However, studies in adolescents with DS are limited, mainly based on cross-sectional designs and with slightly different results [12,13,14]. For example, Izquierdo-Gomez et al. [12] showed that there was a positively association between total PA or VPA and cardiorespiratory fitness. Similarly, Matute-Llorente et al. [13] found evidence that engaging more time in MPA or MVPA were associated with greater levels of cardiorespiratory fitness in this special population. In contrast, Shields et al. [14] found no association between total PA and cardiorespiratory fitness in adolescents with DS. Our results with prospective data showed that those adolescents in the higher tertile of total PA at baseline performed a worsened their cardiorespiratory fitness at 2-year follow-up.

In order to contribute to reduce the deleterious effects of low levels of health-related physical fitness, some studies have examined the influence of exercise programs in each component of health-related physical fitness in adolescents with and without disability. In adolescents without disability, the scientific literature showed that higher PA program attendance was associated with improvements in muscular and cardiorespiratory fitness [28,29]. Focusing on adolescents with DS, different literature review regarding the effect of training on health-related physical fitness showed inconclusive results [30,31,32]. For instance, Gonzalez-Agüero et al. [31] showed that adaptations had not been achieved in cardiovascular fitness when mild aerobic training is performed. Later, Pitetti et al. [32] showed that the adolescent population with DS was able to improve their levels of cardiorespiratory fitness and motor fitness after performing aerobic programs. However, Li et al. [30] showed a trend of improving muscular strength but other outcomes as cardiorespiratory fitness showed less conclusive evidence. These results confirm the need to generate a greater number of studies into DS population with the aim of concretely understanding the effect of training programs on the cardiorespiratory and muscular fitness of this population.

The discordant results for the present study could be partially explained by the lower levels of PA, specially, in DS group [33]. It may be possible that low levels of PA are not enough to generate change in health-related physical fitness at 2-years follow-up. The discordant results of this study with other found in the scientific literature could be due to the lack of used similarly methodology in cross-sectional and longitudinal studies. This similarity in the methodological criteria could improve to understand the data and to find a solution to the low levels of PA and health-related physical fitness of the adolescent population with DS. Clinical evidence has indicated that regular exercise benefits in health status in people with DS with respect to improving their body composition, aerobic capacity, muscle strength, proprioception, and postural stability. The benefits of increased aerobic work capacity and body composition help reduce the cardiometabolic risk profile of this population. However, more randomized controlled trials of longer duration than the present study are needed to determine the dose-response to exercise and to validate these preliminary empirical findings [34]. One possible solution to the results obtained in the sample of adolescents with SD could be the development of physical activity programs that target neuromuscular deficits before young people re-deal with exercise as with the non-disabled population [35]. These programmes should focus on the development of strength, endurance, and FMS at an early age as these components are now understood to play an important role in the development of other capacities such as cardiorespiratory, agility or balance [34]. It has even been shown that strength can play an important role throughout the lives of people with DS as low levels of strength have been associated in lower extremities with an increased risk of falls [36]. Similar relationships have been observed as in strength showing positive associations between cardiorespiratory capacity and balance [37] or strength [38]. In addition to these components, we cannot forget the importance of FMS at an early age. Research in recent decades has shown that FMS proficiency is important for promoting physical (i.e., cardiorespiratory fitness, he althy weight) [39]. In children with DS the development of FMS is delayed, which has been suggested to be associated with balance deficits. The importance of FMS proficiency in child development has been well established in children with typical development but not in DS population, and we are now at a stage where efforts need to be directed at ensuring that children of all abilities are equipped with these skills [40]. Finally, children with DS demonstrate vestibular, sensory, motor and perceptual impairments which manifests as decreased levels of balance, strength, and motor coordination. Together these issues may decrease functional ability leading to more sedentary lifestyles. Use of vestibular stimulation therapy has been attempted to assist in improving motor control and balance in this population with positive results. In summary, it is necessary to develop physical activity programs aimed at improving strength, endurance, balance and FMS in the child population with DS, which will have an impact on health improvement. The strengths of this study are the prospective design with a relatively large sample of adolescents with DS, and considering a sex-matched sub-sample of adolescents without DS for comparative purposes. We used accelerometry to accurately assess PA levels, and health-related physical fitness was evaluated using a standardized battery for adolescents with and without DS. We controlled the analysis with different potential confounders, although there are other variables that we could not take into account (i.e., genetic, dietary patterns, sleep). This study also presents some limitations. The amount of PA may be underestimated because the accelerometers used in the present study are not able to assess swimming or aquatics activities, so levels of PA may have been underestimated. The sample of non-DS could only be “sex-mached” and not “age-mached”, which makes the comparison less robust. The absence of standardized cut points for the population with DS suggests that the results should be interpreted with caution. The sample size of adolescents with DS is relatively small, and the reported results are not representative or generalizable. Finally, the study could not control the variables of caloric intake, lipids, or blood pressure, which could affect the results obtained [41].

## 5. Conclusions

In conclusion, our findings showed that, in adolescents with DS: (i) PA level was not prospectively associated with muscular fitness; (ii) high levels of total PA at baseline were inversely associated with motor and cardiorespiratory fitness at a 2-year follow-up. In adolescents without DS, (iii) those with high levels of MPA showed an improvement in muscular fitness; (iv) only those in the middle tertile of MVPA were associated with an increase of cardiorespiratory fitness at a 2-year follow-up. These results allow us to understand the impact of PA on health-related physical fitness in adolescents with DS. The importance of physical exercise programs in early development ages in a supervised and periodized manner is evident with the aim of improving the physical condition of children with DS. Since this study obtained uncertain results, more longitudinal studies are required in order to verify and compare our results.

## Figures and Tables

**Table 1 ijerph-18-05521-t001:** Descriptive characteristics of study samples at baseline and follow-up.

	*n*	Baseline	2-Years Follow-Up	Change	*p*
**Down Syndrome**					
Age (years)	92	16.0 ± 2.3	17.9 ± 2.3	1.9 ± 0.2	**<0.001**
Weight (kg)	92	53.1 ± 11.6	57.1 ± 11.4	4.1 ± 4.5	**<0.001**
Height (cm)	92	148.6 ± 9.3	151.0± 8.5	2.4 ± 4.1	**<0.001**
Body mass index (kg/m^2^)	92	23.8 ± 4.1	25.0 ± 4.2	1.2 ± 1.7	**<0.001**
Muscular fitness (Z-score) ^a^	92	0.1 ± 1.8	−0.1 ± 1.7	−2.5 ± 19.8	0.358
Motor fitness (sec x−1) ^b^	92	−19.3 ± 4.6	−19.0 ± 4.5	0.4 ± 2.8	0.325
CRF (laps)	92	8.4 ± 6.1	10.1 ± 6.9	1.6 ± 4.6	**<0.001**
Total PA (cpm)	92	331.6 ± 150.2	342.4 ± 117.8	−11.2 ± 122.2	0.983
Moderate PA (min/day)	92	33.7 ± 11.7	31.8 ± 10.4	−14.2 ± 17.1	0.070
Vigorous PA (min/day)	92	12.1 ± 10.0	11.6 ± 9.7	−0.5 ± 7.2	0.564
MVPA (min/day)	92	45.8 ± 19.3	43.3 ± 17.1	−2.6 ± 15.3	0.123
**Non-Down syndrome**					
Age (years)	155	13.8 ± 1.4	15.8 ± 1.4	2.0 ± 0.1	**<0.001**
Weight (kg)	155	55.2 ± 11.9	63.2 ± 11.3	8.0 ± 6.6	**<0.001**
Height (cm)	155	162.0 ± 9.4	169.8 ± 7.9	7.9 ± 5.4	**<0.001**
Body mass index (kg/m^2^)	155	20.9 ± 3.3	21.8 ± 3.2	0.9 ± 2.1	**<0.001**
Muscular fitness (Z-score) ^a^	155	0.1 ± 1.9	0.1 ± 1.7	0.1 ± 1.1	0.150
Motor fitness (sec x−1) ^b^	155	−11.9 ± 1.1	−11.5 ± 1.2	0.4 ± 0.9	**<0.001**
CRF (laps)	155	51.9 ± 23.6	60.0 ± 26.8	8.2 ± 19.3	**<0.001**
Total PA (cpm)	155	471.7 ± 396.0	358.2 ± 203.3	−113.9 ± 248.3	**<0.001**
Moderate PA (min/day)	155	43.1 ± 13.1	39.4 ± 15.8	−3.6 ± 14.7	**0.006**
Vigorous PA (min/day)	155	24.6 ± 14.2	24.9 ± 14.8	0.7 ± 13.1	0.542
MVPA (min/day)	155	67.7 ± 23.7	64.4 ± 25.5	−2.9 ± 22.9	0.147

Values are mean ± SD. Abbreviations: cpm: counts per minute; CRF: cardiorespiratory fitness; PA: physical activity; MVPA: moderate to vigorous PA; *p*: significant differences between baseline and 2-year follow-up (paired *t*-test). ^a^ Z-score computed from handgrip strength and standing broad jump test. ^b^ This score was reverted since lower values in the 4 × 10 m test indicate better performance. Statistically significant values are in bold (*p* < 0.05).

**Table 2 ijerph-18-05521-t002:** Associations of tertiles of physical activity at baseline and health-related physical fitness at 2-year follow-up in adolescents with and without Down syndrome.

		Down Syndrome				NON-Down Syndrome		
		Muscular Fitness (Z-Score) ^a^	Motor Fitness (sec x−1) ^b^	CRF (laps)		Muscular Fitness (Z-Score) ^a^	Motor Fitness (sec x−1) ^b^	CRF (laps)
**Model 1**	*n*	B−Coefficient (95% CI)	B−Coefficient (95% CI)	B−Coefficient (95% CI)	*n*	B−Coefficient (95% CI)	B−Coefficient (95% CI)	B−Coefficient (95% CI)
Total PA (counts/min)								
Low	34	Reference	Reference	Reference	45	Reference	Reference	Reference
Middle	29	−0.49 (−1.26; 0.28)	−0.84 (−3.17; 1.44)	−2.50 (−5.79; 0.78)	55	−0.30 (−0.79; 0.19)	0.05 (−0.30; 0.41)	7.06 (−0.93; 15.05)
High	29	−0.32 (−1.10; 0.46)	−1.24 (−3.54; 1.05)	−1.80 (−5.12; 1.50)	55	0.44 (−0.09; 0.96)	**0.45 (0.07; 0.84)**	**14.08 (5.57; 22.59)**
Moderate PA (min/day)								
Low	29	Reference	Reference	Reference	50	Reference	Reference	Reference
Middle	33	0.34 (−0.44; 1.13)	1.41 (−0.88; 3.71)	1.52 (−1.83; 4.87)	51	−0.32 (−0.81; 0.16)	−0.11 (−0.47; 0.25)	−1.11 (−9.15; 6.93)
High	30	0.33 (−0.48; 1.14)	0.93 (−1.41; 3.27)	0.77 (−2.65; 4.20)	54	0.31 (0.19; 0.81)	0.10 (−0.26; 0.47)	4.91 (−3.30; 13.13)
Vigorous PA (min/day)								
Low	29	Reference	Reference	Reference	47	Reference	Reference	Reference
Middle	33	0.77 (−0.07; 1.50)	1.10 (−1.18; 3.39)	1.21 (−2.01; 4.45)	56	0.14 (−0.38; 0.68)	0.19 (−0.20; 0.57)	5.71 (−2.73; 14.16)
High	30	0.68 (−0.13; 1.49)	**2.80 (0.45; 5.15)**	**5.23 (1.91; 8.55)**	52	**0.63 (0.08; 1.19)**	**0.45 (0.04; 0.85)**	**15.23 (6.37; 24.09)**
MVPA (min/day)								
Low	29	Reference	Reference	Reference	48	Reference	Reference	Reference
Middle	33	0.33 (−0.47; 1.13)	0.98 (−1.36; 3.23)	3.21 (−0.14; 6.65)	56	−0.18 (−0.69; 0.31)	0.33 (−0.28; 0.70)	**11.81 (3.80; 19.83)**
High	30	0.18 (−0.63; 0.99)	1.16 (−1.21; 3.53)	2.27 (−1.12; 5.67)	51	0.38 (−0.15; 0.91)	0.29 (−0.09; 0.68)	**11.94 (3.42; 20.46)**
**Model 2**								
Total PA (counts/min)								
Low	34	Reference	Reference	Reference	45	Reference	Reference	Reference
Middle	29	−0.37 (−0.88; 0.14)	−0.29 (−1.70; 1.13)	−1.46 (−3.67; 0.74)	55	−0.02 (−0.34; 0.30)	0.08 (−0.21; 0.37)	3.12 (−3.37; 9.58)
High	29	−0.29 (−0.79; 0.22)	**−1.46 (−2.88; −0.05)**	**−2.22 (−4.42; 0.02)**	55	0.24 (−0.10; 0.58)	0.21 (−0.10; 0.52)	2.20 (−4.92; 9.32)
Moderate PA (min/day)								
Low	29	Reference	Reference	Reference	50	Reference	Reference	Reference
Middle	33	0.13 (−0.38; 0.64)	0.84 (−0.59; 2.28)	0.31 (−1.94; 2.57)	51	−0.02 (−0.33; 0.28)	−0.07 (−0.36; 0.21)	−0.33 (−6.54; 5.87)
High	30	−0.11 (−0.66; 0.43)	−0.02 (−1.48; 1.49)	−0.77 (−3.12; 1.58)	54	**0.44 (0.13; 0.75)**	0.16 (−0.12; 0.45)	2.92 (−3.34; 9.18)
Vigorous PA (min/day)								
Low	29	Reference	Reference	Reference	47	Reference	Reference	Reference
Middle	33	−0.16 (−0.72; 0.40)	0.04 (−1.48; 1.49)	0.05 (−2.23; 2.34)	56	0.14 (−0.20; 0.48)	0.22 (−0.08; 0.52)	2.96 (−3.71; 9.62)
High	30	−0.34 (−0.92; 0.24)	0.40 (−1.17; 1.99)	1.66 (−0.84; 4.16)	52	0.19 (−0.17; 0.55)	0.17 (−0.15; 0.50)	2.46 (−4.97; 9.89)
MVPA (min/day)								
Low	29	Reference	Reference	Reference	48	Reference	Reference	Reference
Middle	33	0.16 (−0.35; 0.68)	0.45 (−1.02; 1.92)	1.13 (−1.20; 3.46)	56	0.03 (−0.28; 0.35)	0.26 (−0.02; 0.55)	**6.39 (0.07; 12.70)**
High	30	−0.20 (−0.74; 0.33)	−0.02 (−1.52; 1.51)	0.21 (−2.19; 2.61)	51	0.32 (−0.02; 0.65)	0.15 (−0.16; 0.45)	4.95 (−1.79; 11.69)

Values are nonstandardized regression beta coefficients (95% CI). Abbreviations: CRF: cardiorespiratory fitness; PA: physical activity; MVPA: moderate to vigorous PA. Model 1 was adjusted for sex and age (years) at baseline. Model 2 was additionally adjusted for BMI (kg/m^2^) and health-related physical fitness component at baseline, as appropriate; BMI (kg/m^2^) at baseline, the baseline measure of physical fitness, and adaptive behavior was included as a covariate for Down syndrome group. ^a^ Z-score computed from handgrip strength and standing broad jump test. ^b^ This score was reverted since lower values in the 4 × 10 m test indicate better performance. Statistically significant values are in bold (*p* < 0.05).

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
