# Peer review of "Prospective Associations of Physical Activity and Health-Related Physical Fitness in Adolescents with Down Syndrome: The UP&DOWN Longitudinal Study"

_ijerph, 2021, doi:10.3390/ijerph18115521_

Round 1

Reviewer 1 Report

Interesting study and novel findings, but I have some concerns.

Line 22: define DS, although I know it is down syndrome.

Line 25: (2) for method section, but we did not see (1) for background.

Line 29 and 30: B, spell out, Beta.

Line 70: did you consider adding this study aim in the abstract section?

Line 83: it might not proper to say control group, as this was not an experimental study.

Line 86: how many special education schools did you recruit? Please report it.

Line 91: purposed? A spelling error?

Line 170: linear regression used? Why not mixed linear regression?

Line 214-223: Research findings were truly based on results and of values.

Line 230-233: did you consider explaining the lines? It would be more valuable.

Author Response

Response to the Reviewer’s Comments

Reviewer #1

We appreciate the comments and suggestions made which have certainly served to improved the manuscript. Please, see below for a detailed response to each comments as well as a description of the changes made in the manuscript.

Reviewer #1 (Comments to the Author):

Line 22: define DS, although I know it is down syndrome.

Line 25: (2) for method section, but we did not see (1) for background.

Line 29 and 30: B, spell out, Beta.

Line 70: did you consider adding this study aim in the abstract section?

Line 83: it might not proper to say control group, as this was not an experimental study.

Line 86: how many special education schools did you recruit? Please report it.

Line 91: purposed? A spelling error?

Line 214-223: Research findings were truly based on results and of values.

Authors’ response

We truly appreciate al your comments. We have considered all the following points.

Comment

Line 170: linear regression used? Why not mixed linear regression?

 Authors’ response

 This study has chosen the use of simple linear regression statistics for three main reasons. These reasons would be:

Taking into account the values obtained during the analysis of the adjusted coefficient of determination (adjusted R squared) is the measure that defines the percentage explained by the variance of the regression in relation to the variance of the explained variable. That is, the same as the R squared, but with a difference, the adjusted coefficient of determination penalizes the inclusion of variables. Models that have best-fitted and predicted R-squared values are generally chosen. These statistics are designed to avoid a key problem with regular R-squared: it increases every time a predictor is added and can be misleading into specifying an overly complex model. The adjusted R-squared increases only if the new term improves the model more than would be expected by chance, and may also decrease with poor-quality predictors. R 2 is considered  a measure that contains bias. As more variables are incorporated, R 2  tends to overestimate the predictive power of the model, even though the variables are not significant. The R 2 adjust  corrects this interpretation and penalizes the incorporation of new variables

Another reason why a simple linear regression is used is due to the use of a single independent variable in the statistic (physical activity).

Finally, we attend to the data observed in linear regression on statistic F.The F statistic is a test used to evaluate the explanatory power of a group of independent variables on the variation of the dependent variable. In this way, the F statistic aims to determine if, among a group of independent variables, at least one has the capacity to explain a significant part of the variation of the dependent variable. This test is commonly used in statistical inference to carry out hypothesis tests . The result of its calculation is compared with the critical value of a Snedecor F distribution with the confidence level assigned for the test. It is mainly used in multiple regressions although it can also be used in simple regression obtaining the same conclusion as using another statistic called the t statistic.

Comment

Line 230-233: did you consider explaining the lines? It would be more valuable.

Authors’ response

We truly appreciate your comments. We included extra information in the manuscript.

Reviewer 2 Report

Dear editor and authors, thank you very much for the opportunity to review this manuscript.

This paper presents new interesting data on the prospective relationships between PA and physical fitness in a group of children with DS.

The paper is quite original, well written, the statistical analyses is appropriate as well as the results are clearly described. However, despite these strengths, I have some theoretical concerns/questions that I tried to summarize in the following comments for authors, and that limit the contribution of the study to the field. I hope the authors will find these comments helpful. Consequently, in my opinion, the manuscript should be considered for publication in the journal after review these issues in the discussion section.

Comments for authors:

The manuscript is powerful, however, there are 2/3 ideas that I reckon could be interesting to introduce in the discussion section:

1 – The results of model I and changes produced in physical fitness for those participants with high levels of PA point out the importance of PA in motor fitness and CRF in DS individuals. Could indicate these results the special and important role that developing strength could have in these individuals?

As some authors recently explain in children without disabilities, since a prerequisite level of muscular strength is needed to jump, kick, throw, run or doing other activities, concrete efforts are needed to “activate” this generation of children with developmentally appropriate interventions that target neuromuscular deficits before youth become resistant to exercise and sport programs (1). Resistance training programs have showed their benefits in this population (2-3).

A deeper discussion of this point would improve significantly the quality of the paper. From my point of view, this is a beautiful idea that authors must add in the paper due to the relevance of its manuscript and their last works.

2 – Line 259. The controversial associations found also evidence the significance not only the quantity of total physical activity but also the start -in the early stages of development- in physical exercise programs. Periodizing and supervising specific intervention for DS children will be key in order to improve their physical fitness. Moreover, the quality of movement/interventions is more important than other items (4). This is an important statement for sport sciences professionals.

3 – Line 278-280. I am not sure about that statement. I think is important to underline the role of physical exercise again here and cancel this phrase.

4 – Introduce in the conclusions section some of these ideas.

Other minor corrections:

  • Line 5: Only the first letter of name and surname in capital letters.
  • Line 18: Provide the detailed correspondence (name and tel.).
  • Line 34: From my point of view, authors should change the keywords because these words appear in the title too. Keywords are important because they are the linchpin between what people are searching for and the content you are providing to fill that need. Please find other keywords related with the main research issue in order to offer the possibility find this article easily to other authors.
  • Line 88: Change should accomplished by “had to accomplish”.
  • Line 90: Change disability by motor disability.
  • Line 106: How many times performed the handgrip test with each hand the participants? This must be clarified.
  • Line 152: Provide the information about the city and country of the instrument of measure.
  • Line 159: “…internal consistency ( =0.87)…” à Provide the sign.
  • Please check the manuscript because there are lines in it in which there are extra spaces between words. Line 57, line 60, line 129, line 132 and line 207 are only few examples.

The paper is well presented and I recommend it for publication after the previous suggestions. Thank you again to the editor and authors for the possibility to review this manuscript. Congratulations also to the authors for the UP&DOWN project. I have learnt a lot so it was a pleasure.

References:

  1. Faigenbaum, A. D., & Geisler, S. (2021). The Promise of Youth Resistance Training. B&G Bewegungstherapie und Gesundheitssport37(02), 47-51.
  2. Sayadinezhad, T., & Abdolvahab, M., & Akbarfahimi, M., & Jalili, M., & Rafiee, S., & Baghestani, A. (2013). The study of the effect of progressive resistance training on functional balance of 8-12 years old children with down syndrome. Journal of modern rehabilitation, 7(1), 29-33.
  3. Shields, N., & Taylor, N. F. (2010). A student-led progressive resistance training program increases lower limb muscle strength in adolescents with Down syndrome: a randomised controlled trial. Journal of physiotherapy56(3), 187-193.
  4. Moreno-Garcia, G., Monteagudo-Chiner, P., & Cabedo-Mas, A. (2020). The role of music in the development of children with Down syndrome: a systematic review. Interdisciplinary Science Reviews45(2), 158-173.

Author Response

Response to the Reviewer’s Comments

Reviewer #2

We appreciate the comments and suggestions made which have certainly served to improved the manuscript. Please, see below for a detailed response to each comments as well as a description of the changes made in the manuscript.

Reviewer #2 (Comments to the Author):

Other minor corrections:

Line 5: Only the first letter of name and surname in capital letters.

Line 18: Provide the detailed correspondence (name and tel.).

Line 34: From my point of view, authors should change the keywords because these words appear in the title too. Keywords are important because they are the linchpin between what people are searching for and the content you are providing to fill that need. Please find other keywords related with the main research issue in order to offer the possibility find this article easily to other authors.

Line 88: Change should accomplished by “had to accomplish”.

Line 90: Change disability by motor disability.

Line 106: How many times performed the handgrip test with each hand the participants? This must be clarified.

Line 152: Provide the information about the city and country of the instrument of measure.

Line 159: “…internal consistency ( =0.87)…” à Provide the sign.

Please check the manuscript because there are lines in it in which there are extra spaces between words. Line 57, line 60, line 129, line 132 and line 207 are only few examples.

1 – The results of model I and changes produced in physical fitness for those participants with high levels of PA point out the importance of PA in motor fitness and CRF in DS individuals. Could indicate these results the special and important role that developing strength could have in these individuals?

As some authors recently explain in children without disabilities, since a prerequisite level of muscular strength is needed to jump, kick, throw, run or doing other activities, concrete efforts are needed to “activate” this generation of children with developmentally appropriate interventions that target neuromuscular deficits before youth become resistant to exercise and sport programs (1). Resistance training programs have showed their benefits in this population (2-3).

A deeper discussion of this point would improve significantly the quality of the paper. From my point of view, this is a beautiful idea that authors must add in the paper due to the relevance of its manuscript and their last works. 

2 – Line 259. The controversial associations found also evidence the significance not only the quantity of total physical activity but also the start -in the early stages of development- in physical exercise programs. Periodizing and supervising specific intervention for DS children will be key in order to improve their physical fitness. Moreover, the quality of movement/interventions is more important than other items (4). This is an important statement for sport sciences professionals.

3 – Line 278-280. I am not sure about that statement. I think is important to underline the role of physical exercise again here and cancel this phrase.

Authors’ response

We included this information in the manuscript.

Reviewer 3 Report

Proposed paper is interesting and well written. However, some revision are needed before it can be accepted for pubblication:

  • Are down subjects free of any heart malformation? please report?
  • Are data on BP and lipids control available? if yes report, if no state it as a limitation of the work.
  • What about diet and it's eventually variation during the follow-up. if data are available report it and include it into the multivariate model, if no state it as a limitation of the work. On this point please cite also the following: Nutrients. 2021 Feb 16;13(2):634.
  • Health behaviour at follow-up was associated with physical activity at baseline. Are data on physical activity at follow-up available? if yes please done specific analysis also for this, if not report in the limitation section.

Author Response

Response to the Reviewer’s Comments

Reviewer #3

We appreciate the comments and suggestions made which have certainly served to improved the manuscript. Please, see below for a detailed response to each comments as well as a description of the changes made in the manuscript.

Reviewer #3 (Comments to the Author):

Are down subjects free of any heart malformation? please report?

Are data on BP and lipids control available? if yes report, if no state it as a limitation of the work.

What about diet and it's eventually variation during the follow-up. if data are available report it and include it into the multivariate model, if no state it as a limitation of the work.

Authors’ response

We truly appreciate al your comments. We have considered all the following points.

Comment

Health behaviour at follow-up was associated with physical activity at baseline. Are data on physical activity at follow-up available? if yes please done specific analysis also for this, if not report in the limitation section.

Authors’ response

One of the possible limitations that this study may have focuses on the assessment of the association of health behavior and physical activity at follow-up. As you can see from the manuscript, this work belongs to  UP&DOWN study research group. In it we participate different colleagues with the intention of showing the findings produced in our research. The drawback we have in this case is the absence of physical activity at follow-up because they belong to another study conducted with population with DS. This research is in the development phase so the data of the physical activity at follow-up cannot be shown in both works. While true, the analyses you recommend in the review were done to get a slight idea of what would happen. That's certainly a really interesting analysis to understand some important aspects. The data obtained in the analysis show very similar results to those shown in this study. That is, physical activity at follow-up does not seem to show an association with the fitness variable in this population. I hope and wish to have clarified your doubts about it. Thank you very much for this interesting comment.

Round 2

Reviewer 3 Report

Authors replies to all the query raised and paper improves and can now be accepted for pubblication.